# Characterization of Physical and Biological Properties of a Caries-Arresting Liquid Containing Copper Doped Bioglass Nanoparticles

**DOI:** 10.3390/pharmaceutics14061137

**Published:** 2022-05-27

**Authors:** Se-Jung Bang, Soo-Kyung Jun, Yu-Jin Kim, Jun-Yong Ahn, Huong Thu Vu, Nandin Mandakhbayar, Mi-Ran Han, Jun-Haeng Lee, Jong-Bin Kim, Jong-Soo Kim, Jonathan C. Knowles, Hye-Sung Kim, Hae-Hyoung Lee, Ji-Sun Shin, Jung-Hwan Lee

**Affiliations:** 1Department of Pediatric Dentistry, College of Dentistry, Dankook University, Cheonan 31116, Korea; tinybunny97@naver.com (S.-J.B.); miraneee@dankook.ac.kr (M.-R.H.); haeng119@naver.com (J.-H.L.); jbkim0222@dankook.ac.kr (J.-B.K.); jskim@dku.edu (J.-S.K.); 2Department of Biomaterials Science, College of Dentistry, Dankook University, Cheonan 31116, Korea; iris979@hanmail.net (S.-K.J.); yujin10316426@gmail.com (Y.-J.K.); junyoung@gmail.com (J.-Y.A.); hskim1213@dankook.ac.kr (H.-S.K.); haelee@dku.edu (H.-H.L.); 3Institute of Tissue Regeneration Engineering (ITREN), Dankook University, Cheonan 31116, Korea; huong.vuthudr@gmail.com (H.T.V.); m.nandia@gmail.com (N.M.); 4Department of Dental Hygiene, Hanseo University, Seosan 31962, Korea; 5Department of Nanobiomedical Science & BK21 PLUS NBM Global Research Center for Regenerative Medicine, Dankook University, Cheonan 31116, Korea; j.knowles@ucl.ac.uk; 6UCL Eastman-Korea Dental Medicine Innovation Centre, Dankook University, Cheonan 31116, Korea; 7Cell & Matter Institute, Dankook University, Cheonan 31116, Korea; 8Division of Biomaterials and Tissue Engineering, Eastman Dental Institute, Royal Free Hospital, Rowland Hill Street, London NW3 2PF, UK; 9Department of Regenerative Dental Medicine, School of Dentistry, Dankook University, Cheonan 31116, Korea; 10Mechanobiology Dental Medicine Research Center, Cheonan 31116, Korea

**Keywords:** silver diamine fluoride, copper-doped bioactive glass nanoparticle, hydroxyapatite disc, viscosity, physicochemical properties, biological properties, pulp stem cells, *Streptococcus mutans*, *Staphylococcus aureus*

## Abstract

Silver diamine fluoride (SDF) is an outstanding dental material for arresting and preventing caries, but some drawbacks, such as high flowability due to low viscosity and cytotoxicity to the pulp, have been reported. To overcome these problems, copper-doped bioactive glass nanoparticles (CuBGns) were combined with SDF. After synthesis, CuBGns were examined by physical analysis and added in SDF at different weight/volume% (SDF@CuBGn). After assessing physical properties (viscosity and flowability) of SDF@CuBGn, physicochemical properties (morphology before and after simulated body fluid (SBF) immersion and ion release) of SDF@CuBGn-applied hydroxyapatite (HA) discs were evaluated. Biological properties were further evaluated by cytotoxicity test to pulp stem cells and antibacterial effect on cariogenic organisms (*Streptococcus mutans* and *Staphylococcus aureus*). Combining CuBGns in SDF increased the viscosity up to 3 times while lowering the flowability. More CuBGns and functional elements in SDF (Ag and F) were deposited on the HA substrate, even after SBF immersion test for 14 days, and they showed higher Cu, Ca, and Si release without changing F and Ag release. Cell viability test suggested lower cytotoxicity in SDF@CuBGn-applied HA, while CuBGns in SDF boosted antibacterial effect against *S. aureus*, ~27% in diameter of agar diffusion test. In conclusion, the addition of CuBGn to SDF enhances viscosity, Ag and F deposition, and antibacterial effects while reducing cell toxicity, highlighting the role of bioactive CuBGns for regulating physical and biological effects of dental materials.

## 1. Introduction

In the past, dental caries was recognized as a progressive disease that eventually destroys teeth and therefore requires surgical or restorative intervention [1]. However, it has been revealed that, first, restorative treatment alone cannot stop the progression of caries [2], second, there is no permanent restoration, and, lastly, some caries may not progress. Therefore, the treatment of dental caries is moving toward identifying the risk of each patient and managing disease progression as an appropriate preventative treatment supplemented with restorative treatment if necessary. Rather than removing caries and restoring the hole, the initial caries lesions can be managed by preserving the tooth structure through nonoperative treatment inducing remineralization, mostly using fluoride [3].

Silver nitrate is a well-known antimicrobial substance that has been used in medicine and dentistry since the 1800s [4]. By combining silver nitrate with fluoride, which has a caries prevention effect, silver diamine fluoride (SDF) has been used for arresting and preventing caries in Japan since the 1960s, and it is a very effective and cost-efficient method among known caries prevention methods [5]. It can be useful approach to deal with some caries for which conventional dental restorative treatment cannot be performed, such as a young patient with severe dental fear and anxiety [6]. However, as a major disadvantage of SDF, it causes black staining when the silver ions are exposed to light [7]. Second, since it is a highly flowable liquid similar to water, it may flow into unwanted areas even if the caries is spot treated topically. This may cause SDF to be diluted in saliva, which results in a lower concentration of applied fluoride [8], and gingivitis can occur when SDF comes into contact with the gingiva [9]. Although there have not been sufficient experiments on the cytotoxicity of SDF, it was shown to be toxic to fibroblasts for up to 9 weeks, even after rinsing [10]. Additionally, it was reported that SDF can cause pulp necrosis when it comes into contact with pulp [11], so it seems to be cytotoxic to pulp cells [7]. To overcome these shortcomings of SDF, various attempts have been made, such as using potassium iodide (KI) to reduce staining by removing excess silver ions [12] and applying fluoride varnish on SDF to increase the contact time between SDF and the tooth surface [13].

Bioactive glasses (BGs) were first fabricated by Hench et al. in 1969 as a promising material for tissue engineering and regeneration [14]. The utilization of BGs as a dental material began in the mid-1980s, when BGs were successfully used to preserve the alveolar bone of edentulous patients [15]. Furthermore, BGs have been developed into bioactive glass nanoparticles (BGns), which have a high surface area, bioactivity, and antibacterial effect [16,17]. Many researchers have attempted to mix BGns with various materials and have reported improvements in mechanical properties, bioactivity, and mineralization capacities [18,19,20,21]. In addition, incorporating therapeutic metallic ions can enhance osteogenesis, angiogenesis, and antimicrobial effects [22,23,24,25,26,27]. Notably, as a therapeutic ion, copper has a strong antimicrobial effect over a wide range but low toxicity for humans [28]. Therefore, a favorable effect can be expected by mixing copper-doped BGn (CuBGn) with other dental materials; for example, Choe et al. revealed that ZPC mixed with CuBGn showed relatively good in vitro cell viability and antibacterial effects without affecting mechanical properties [29].

In this study, CuBGns were mixed within SDF, and their physical properties, such as viscosity and flowability, were compared. Additionally, to simulate the oral environment, hydroxyapatite (HA) discs were used to evaluate physicochemical properties, such as released ion analysis and surface analysis as well as biological properties, including cell toxicity against pulp stem cells and antibacterial effects on cariogenic organisms. The null hypothesis is that there are no differences in the (1) physical properties, (2) physicochemical properties, and (3) biological effects between bare SDF and SDF combined with CuBGn at several concentrations.

## 2. Materials and Methods

### 2.1. Preparation and Characterization of CuBGns

Briefly, the alkali-mediated sol–gel synthesis method was used to prepare CuBGns. Copper nitrate and calcium nitrate tetrahydrate were used to include Cu in the silicate glass network in bioactive glass nanoparticles. First, hexadecyltrimethylammonium bromide (CTAB; Sigma Aldrich, St. Lois, MO, USA) was liquefied in distilled water (DW). Then, ethyl alcohol anhydrous (C_2_H_5_OH; C_2_H_6_O, 99.5%, Daejung, Siheung-si, Republic of Korea), ammonium hydroxide (NH_4_OH, 28.0% NH_3_ in water ≥99.99% metal basis), ethyl ether (Daejung), tetra-hydrate calcium nitrate (Sigma Aldrich), tetraethyl orthosilicate (TEOS; Sigma Aldrich), and copper nitrate were added. The ratio of Ca:Si:Cu for CuBGns was set at 10:85:5 in wt%. The mixed solution was agitated at room temperature for 4 h and precipitated with acetone. Thereafter, the mixture was centrifuged at 10,000 rpm for 3 min, and CuBGn was obtained. The powder was dried overnight at 60 °C in an oven and baked at 550 °C for 10 h to remove the remaining CTAB and produce a stable glass network [30].

The morphology of CuBGns was examined using field emission scanning electron microscopy (FE-SEM; Sigma 300, ZEISS, Gottingen, Germany) and analyzed by energy dispersive spectroscopy (EDS; Noran System Seven, Thermo Fisher Scientific, Waltham, MA, USA) to detect the elements (50,000×, 5 kV). X-ray diffraction (XRD; Ultima IV, Rigaku, Tokyo, Japan) was also performed to assess the crystal structure of the powder.

### 2.2. Physical Analysis of SDF@CuBGn (SDF Mixed with CuBGn)

SDF (Saforide^®^, Tokyo Seiyaku Kasei Co. Ltd., Osaka, Japan) was divided into four groups, and CuBGn was added at 0, 1, 5, and 10 (*w/v*)%. Each mixed solution was sonicated and vortexed at room temperature for 1 min and named SDF@CuBGn0, SDF@CuBGn1, SDF@CuBGn5, and SDF@CuBGn10 accordingly.

A discovery HR-1 instrument (TRIOS, TA Instruments, New Castle, DE, USA) was used for the viscosity test. SDF@CuBGn (0.3 mL) was placed evenly between a parallel plate (diameter 60.0 mm), and the gap was set to 100 μm. All analyses were performed at room temperature with a 1~3000 (1/s) shear rate, 20 s duration, and each measurement was repeated three times and averaged (n = 3).

Flow tests were performed according to the method of ISO 6876:2012 (Dentistry—Root canal sealing materials) because there is no ISO standard on the flowability of SDF. Two glass plates were prepared, which are at least 40 mm × 40 mm and 5 mm thick, with a mass of approximately 20 g. In ISO 6876:2012, the mass of the test material was 0.05 ± 0.005 mL, but the test was performed with a smaller amount (0.01 mL) since SDF was much more spreadable than root canal sealing materials. The SDF@CuBGn was placed on the center of one glass, and the other glass plate of the same size and mass was placed centrally on top of the sample. An additional weight was placed on the plate so that the total weight on the sample was 120 ± 2 g. The weight was removed after 10 min, and the maximum and minimum diameters of the sample were measured and recorded if they were within 1 mm of each other, and the test was repeated if they were not. Five measurements for each group were averaged (n = 5).

### 2.3. Physicochemical Analysis of SDF@CuBGn with Hydroxyapatite (HA) Discs

HA discs with a diameter of 9.5 mm and a thickness of 1.8 mm (Clarkson Chromatography, South Williamsport, PA, USA) were prepared for simulating the tooth surface with the same composition, hydroxyapatite. The fluoride ion release test was performed according to ISO 10993-12:2012 (Biological evaluation of medical devices—Part 12: Sample preparation and reference materials), ISO 19448:2018, (Dentistry—Analysis of fluoride concentration in aqueous solutions by use of fluoride ion-selective electrode). According to the manufacturer’s instructions, HA discs were prepared by spreading 10 μL of SDF@CuBGn on the surface and blot drying for 1 min. Next, SDF@CuBGn-applied HA discs were soaked in 0.652 mL of artificial saliva (Pickering Laboratories, Mountain View, California, USA), which was calculated based on ISO 10993-12:2012. The medium was stored at 37 °C, agitated, and changed at 1, 4, and 12 h and 1, 2, 3, 7, and 14 days to test the fluoride concentration. The test was repeated three times for each group, and the amount of released fluoride was accumulated and averaged (n = 3).

Next, the concentration of released ions and pH were measured using the extracted media after 24 h. Ag, Ca, Cu, and Si ions, which were detected in CuBGns, were selected. A pH meter (inoLab pH 7110, WTW, Weilheim, Germany) was used to measure the pH at room temperature (24 °C), and inductively coupled plasma atomic emission spectrometry (ICP–AES; Optima 8300, PerkinElmer, MA, USA) was used to evaluate the Ag Ca, Cu, and Si ion concentrations. All samples were analyzed three times and averaged (n = 3).

For surface analysis, SDF@CuBGn0- and SDF@CuBGn10-applied HA discs were prepared and dried for 24 h. The surfaces were examined by FE-SEM and EDS (50,000×, 20 kV). Then, SDF@CuBGn0- and SDF@CuBGn10-applied HA discs were immersed in simulated body fluid (SBF; Na^+^ (142.0 mM), K^+^ (5.0 mM), Mg^2+^ (1.5 mM), Ca^2+^ (2.5 mM), Cl^−^ (147.8 mM), HCO_3_^−^ (4.2 mM), HPO_4_^2−^ (1.0 mM), SO_4_^2−^ (0.5 mM)) at pH 7.4 and 37 °C, which is optimal condition for accelerating remineralization. After 14 days, the samples were thoroughly washed, and the morphology change was observed and analyzed by FE-SEM and EDS (486×, 5000×, 15 kV).

### 2.4. In Vitro Study of Cytotoxicity on Pulp Stem Cells from Human Exfoliated Deciduous Teeth (SHED) and Antibacterial Effect on Cariogenic Organisms

Pulp stem cells were isolated from vital primary teeth of healthy children after consent was obtained from guardians. Because the permanent mandibular incisor erupted abnormally, the primary incisor was extracted and approved by the Ethical Committee off the Institutional Review Board of Dankook University Dental Hospital (IRB number 2019-10-001). SHED was isolated from pulp using the enzymatic dissociation method, and sorting by flow cytometry to purify SHED was performed, as previously described by Masako Miura et al. [31]. The pulp was separated and then digested in a solution of 2 mg/mL collagenase type I (Worthington Biochemical, Lakewood, NJ, USA) and 4 mg/mL dipase (Invitrogen, Carlsbad, CA, USA) for 1 h at 37 °C in a water bath. The cell suspension was centrifuged at 1500 rpm for 5 min. The cells were cultured in α-minimum essential medium (α-MEM; Gibco BRL, Grand Island, NY, USA) supplemented with 15% heated in active fetal bovine serum (FBS, Corning, NY, USA), 1% penicillin/streptomycin (PS, Gibco, Gaithersburg, MD, USA), 2 mM GlutaMAX, and 1 mM L-ascorbic acid, and incubated in 5% CO_2_ at 37 °C. The culture medium was changed after 48 h of initial incubation and then every 7 days thereafter. After 10 to 14 days, single-cell-derived colonies were collected and subcultured. Cultured SHED in passages lower than 10 passages were used for testing.

A total of 10 μL of SDF@CuBGn was placed on one side of a sterilized HA disc. Compressed air was blown on the sample at a constant pressure, allowing the solution to flow from one side to the other and the excess solution to fall into α-MEM. After incubating the medium at 37 °C while agitating for 1 h, the media was extracted and filtered with a 0.22 µm filter (Corning) to eliminate contamination. For use in the cytotoxicity assay, the extracted solution was diluted at different ratios (100%, 50%, 25%, 12.5%, and 6.25%) in α-MEM, and then diluted medium was prepared with the addition of 10% FBS and 1% PS.

The toxicity of SDF@CuBGn on SHED was based on live cell viability using live/dead staining (0.5 µM casein AM and 2 µM ethidium homodimer-1 solutions, Thermo Fisher, USA), which determines cell viability in a population based on plasma membrane integrity and esterase activity. In brief, SHED were seeded in a 96-well plate (SPL Life Sciences, Pocheon, Gyeonggi-do, Korea) at a density of 5 × 10^4^ cells/well (α-MEM, 10% FBS and 1% PS) and incubated overnight. Then, the cell culture media was changed to extracted solution at different concentrations for the next 24 h, and the 0% extracted group was used as a negative control. After 24 h, cells with intact cell membranes stained green, whereas cells with damaged membranes appeared red. Cell survival was determined by counting live and dead cells (n = 6).

To measure the antibacterial effect, agar diffusion tests were selected, which are simple, fast, and reliable. As representative cariogenic organisms, *Staphylococcus aureus* (*S. aureus*, ATCC 6538) and *Streptococcus mutans* (*S. mutans*, ATCC 25175) were chosen and purchased from American Type Culture Collection (ATCC; Manassas, VA, USA). *S. aureus* was grown on tryptic soy broth (TSB, Difco Laboratories, Sparks, MD, USA), and *S. mutans* was grown on brain heart infusion (BHI, BD science, Milwaukee, WI, USA) broth. All strains that were grown at 37 °C for 18 h were subcultured separately and diluted using phosphate-buffered saline (PBS; Gibco, Grand Island, NE, USA) to obtain bacterial cell densities of approximately 1 × 10^8^ colony-forming unit (CFU)/mL. Diluted bacterial suspensions were spread with sterilized cotton swabs on TSA and BHI agar plates (100 mm × 15 mm) and allowed to dry for 3–5 min. ADVANTEC paper discs (8 mm diameter, 0.7 mm thickness) that had been sterilized with ethylene gas were placed on TSA and BHI agar plates and pressed to ensure complete contact with the agar surface. Twenty microliters of SDF@CuBGn were loaded onto paper discs, and plates were incubated at 37 °C for 18 h. After incubation, the diameter of the inhibition zones was measured three times and averaged (n = 3).

### 2.5. Statistical Analysis

The data are presented as the mean ± one standard deviation (1 SD). Comparisons among groups were conducted using one-way analysis of variance (ANOVA) and repeated-measures ANOVA only in the fluoride ion release test, with the Tukey post hoc test. The statistical significance level adopted was *p* < 0.05, and SPSS 23.0 (Statistical Package for Social Science, version 23.0, IBM Corporation, Chicago, IL, USA) was used.

## 3. Results

### 3.1. Characterization of CuBGns and Physical Analysis of SDF@CuBGns

CuBGns were fabricated with the sol–gel method, and the morphology and structure of the CuBGns were characterized by FE-SEM, EDS, and XRD (Figure 1B). FE-SEM shows the numerous spherical morphologies of CuBGns, diameter of 254 ± 21 nm, and EDS analysis confirmed the presence and chemical composition of elements Si, Cu, and Ca. Additionally, XRD showed the presence of a broad halo (at 2θ = 20° to 30°) and the absence of XRD diffraction peaks [32].

Figure 1A shows a schematic illustration of the preparation of SDF@CuBGn and an image of the actual mix. CuBGn was placed in SDF for 0, 1, 5, and 10 (*w/v*)%, and the larger the amount of powder was, the bluer the solution.

As more powder was added, the viscosity increased (Figure 1C, *p* < 0.05). SDF@CuBGn10 presented the increased viscosity up to 3 times more than SDF@CuBGn0. Additionally, all the test solutions showed decreasing viscosity as the shear rate increased. Similar to the viscosity, the flow also decreased as the number of particles increased (Figure 1D, *p* < 0.05). SDF@CuBGn0 presented the highest flowability of 35.5 ± 1.5 mm, but SDF@CuBGn10 showed the opposite result of 11.5 ± 1.2 mm.

### 3.2. Physicochemical Analysis of SDF@CuBGn with HA Discs

HA discs were used to simulate when SDF@CuBGn was applied to teeth (Figure 2A,B), and the surface structure was observed with FE-SEM and EDS. While the spherical morphology of numerous CuBGns was distinctively found in discs with SDF@CuBGn10, of course, there was Ag and F in discs with SDF@CuBGn0 but not CuBGns (Figure 2C–F). For the same reason, EDS analysis found Ag and F in both discs, yet Cu and Si were detected in the SDF@CuBGn10 disc only (Figure 2C’,D’,F). After immersing in SBF for 14 days, SDF@CuBGn-applied discs showed many blocks, and EDS analysis revealed they were Ag (Figure 3B,3B’). Particularly, SDF@CuBGn10-applied disc showed that the silver blocks held together to form rod-shaped branches of Ag (Figure 3D’). In SDF@CuBGn10, two spots were picked according to concentration of silver. The spot with lots of silver rods showed high silver content (85.8 ± 13.1 wt%, Figure 3D’), while the other spots showed high fluoride deposition instead of silver (19.3 ± 2.6 wt%, Figure 3C’).

The samples were immersed in artificial saliva to reproduce the oral environment, and the concentration of ions was measured. Figure 4A shows the accumulated result of the fluoride ion release test. Within 4 h, more than 90% of fluoride was released and the ion release rate gradually slowed down. All the test groups showed significantly higher fluoride ion release than the control group by more than 1000 times (*p* < 0.05), but the presence or amount of CuBGn did not affect the amount of released ion or release rate (*p* > 0.05).

In addition, the medium in which HA discs were immersed in artificial saliva for 24 h was subjected to pH analysis and dissolved Ag, Ca, Cu, and Si ion levels by ICP–AES. As the CuBGn powder content increased, the concentration of Ca, Cu, and Si ions in the medium increased (Figure 4B, *p* < 0.05) without affecting the release of Ag (*p* > 0.05), and the pH of the extracted media slightly decreased, although it was higher than that of the control solution (Figure 4C, *p* < 0.05).

### 3.3. In Vitro Study of CuBGn Cytotoxicity on Pulp Stem Cells and Antibacterial Effects on Cariogenic Organisms

The surplus solution on the tooth surface was removed with compressed air and used for the pulp stem cell viability experiment, and as expected, the high-viscosity solution had less surplus solution than the relatively low-viscosity solutions (Figure 5A). Figure 5B shows the fluorescence images of live (green color) and dead (red color) cells, and Figure 5C is a graph generated by counting the number of cells. Surplus test solutions were diluted to 1:1, 1:2, 1:4, 1:8, and 1:16. At 1:1 and 1:2 dilution ratios in culture media, all groups of cells did not survive. At a 1:4 dilution ratio, SDF@CuBGn5 and SDF@CuBGn10 showed better cell viability (approximately 30% compared to the control group) than SDF@CuBGn0 and SDF@CuBGn1 (<1% compared to the control group). At all dilution ratios, as the amount of powder increased, cell viability also increased, and the size of the inhibition zone increased (Figure 6). In *S. mutans* and *S. aureus*, the size of the inhibition zone increased as the amount of powder increased, but it was statistically significant only in *S. aureus* (*p* > 0.05 in *S. mutans*). In *S. aureus*, SDF@CuBGn5 and SDF@CuBGn10 clearly had a higher effect than SDF@CuBGn0 (*p* < 0.05).

## 4. Discussion

The null hypothesis was rejected, except for the fluoride and silver ion release test. Adding CuBGn increased the viscosity (decreased flowability), which led to a decrease in cytotoxicity. CuBGn did not affect the fluoride and silver ion releasing ability of SDF, but increased Ca, Cu, and Si ion release and deposition of Ag and F on HA discs. Moreover, the addition of Cu increased the antibacterial effect.

FE-SEM, EDS, and XRD found the spherical mesoporous morphologies of CuBGns. The previous report with CuBGns revealed broad halo and absence of XRD diffraction peak, which was the same result with this study, and had shown Cu peak in XPS spectrum [30,32].

SDF is an outstanding material for arresting and preventing caries, but side effects such as black staining, high flowability by low viscosity, and cytotoxicity on gingiva and pulp have been reported. To overcome these problems, CuBGn was added to increase the viscosity. Viscosity is a basic property of liquids defined as internal resistance to flow [33]. Low viscosity, which means high flowability, makes the reagent easily reach all the corners, but in other words, it allows it to flow into unwanted areas as well. Thus, nanoparticles were added to SDF in an attempt to increase its viscosity [34]. It was confirmed that the more powder there was, the higher the viscosity (lower the flowability). Additionally, all the test solutions showed thixotropy (decreasing viscosity as the shear rate increases), which means that while the solution is applied to teeth, it has high flowability and spreads well on the intended spot, and flowability decreases after removal of the shear force.

The more CuBGns were added, the higher the concentrations of Ca, Cu, and Si ions were, without changing the F and Ag release (Figure 4A,B), which are the main components of SDF. The extract of SDF@CuBGn medium was expected to be basic since SDF is an alkaline ammonia solution [35]. Interestingly, adding CuBGn made the pH decrease slightly, despite hydrolysis of BGn producing OH^-^ ions [36], and this seems to be due to the amount of SDF@CuBGn attached to the HA disc increasing as the viscosity increases.

Ag and F, the main components of SDF, were detected on the surface of SDF@CuBGn-applied HA discs. Noteworthily, adding CuBGns made silver-branch and even increased fluoride deposition in spots where there was low silver. Because fluoride is involved with remineralization, and silver has an antibacterial effect, when F, Ag, and Cu are detected on the tooth surface, they act in a positive way.

Fluoride at a high concentration (44,600 ppm) in SDF reacts with HA in teeth to produce CaF_2_, which is controversial in terms of its preventive effect [37,38]. Although CaF_2_ does not have as high acid resistance as fluorapatite (Fap), it can generate Fap by reacting with HPO_4_^2−^ during remineralization, thereby serving as a reservoir for fluoride in caries environments [39]. However, Lou et al. showed that SDF with HA forms CaF_2_-like material, which has little therapeutic effect because it is easily washed away by water. Even at low pH, HA and Fap were dissolved on the enamel surface, and CaF_2_ was formed, which was an undesirable result [40]. However, CaF_2_ shows lower solubility in alkaline environments, and because SDF has high alkalinity, a synergistic effect is expected [41,42]. Since the formation and caries prevention effects of CaF_2_ are still controversial, further research is needed.

When SDF was applied to HA, small amounts of Ag_3_PO_4_ [37,38] or nanoscopic metallic silver particles attached to HA were found [40]. Metallic silver (Ag or Ag^0^) is relatively inert, but it can generate silver ions (Ag^+^) by reacting with moisture in the oral cavity. Silver ions have antibacterial properties by inactivating bacterial enzymes, causing leakage in cell membranes, and inhibiting mutagenesis, apoptosis, and cell division by interacting with DNA [43,44,45,46,47]. SBF immersion test suggests the potential for CuBGns to deposit more silver on the tooth surface, which leads to higher antibacterial activity. However, when silver ion is exposed to light, it can make black staining worse. Therefore, it is crucial to consider the effect of light curing on SDF when using under photo-polymerizing restorative materials, such as composite resin. According to Hassan et al., photopolymerization of SDF increases the surface hardness [48]. Quock et al., in addition, reported that SDF have no side effects on the bond strength of resin composite [49]. However, one study from Japan found that SDF affected the tensile bond strength of resin-based adhesive cement [50], which is controversial from other studies. Further research is necessary to figure out these controversial problems.

The presence of copper ions reduces the solubility of enamel. Copper on the tooth surface inhibits demineralization [51], and Sierpinska et al. found that the amount of copper was significantly lower in the enamel of severely worn teeth than in normal teeth [52]. Many researchers applied CuSO_4_ to teeth, and CuSO_4_ significantly diminished the dissolution of enamel [51,53]. Impressively, the combination of fluoride with copper resulted in significantly less caries than fluoride or copper ions alone [54,55,56,57]. In addition, copper has high antimicrobial activity against cariogenic bacteria, consequentially indicating anticaries effects [53,55,56,58]. The core issue in using copper ions as a therapeutic agent for caries prevention is chemical stability, and recent studies have been solving this problem via the use of nanoparticles [59]. Copper nanoparticles show promising effects, such as inhibiting the growth of acidogenic bacteria and biofilms along with high antibacterial properties [60,61,62].

When SDF is applied to the teeth, the solution is applied not only to the target area, but the surplus solution flows to the surrounding tissue, such as gingiva and pulp [8]. SDF that has flowed into the gingiva causes erythema or white lesions on mucosa (which usually disappear within 48 h without treatment [63]), which seems to be because SDF is toxic to human gingival fibroblasts [10]. Additionally, when SDF was applied to deep caries, the microhardness increased from the lesion surface to more than 0.2 mm below the dentin [64], and silver ions of SDF were found in the sound dentine below the carious dentine [65]. This means that the SDF can travel through the dentin tubules deeper than the area applied and therefore closer to the pulp. However, SDF is toxic to pulp cells [66], and cell viability is an important factor, as it directly affects proliferation and differentiation [67]. Therefore, a method was designed to reduce the cell toxicity of SDF. It is hypothesized that if the solution viscosity is increased by mixing powder with SDF, the amount of solution passing through the dentinal tubule is reduced (Hagen–Poiseuille’s law), resulting in better cell viability. In fact, in one experiment, when SDF mixed with zinc oxide powder was applied to dentin just before pulp exposure, no inflammatory cells or necrosis were found in the pulp [68].

At all dilution ratios, the higher the viscosity was, the more solution was attached to the HA disc. Therefore, when compressed air was applied, the amount of solution that could not adhere depended on the amount of powder. As a result, the cytotoxicity decreased as the amount of powder increased.

The antibacterial effect of silver ions is significant to the anticaries effect of SDF just as that of fluorine ions is to remineralization. Silver ions exhibit antibacterial effects by interacting with DNA, inactivating enzymes, and destroying cell membranes, and are also effective against *S. mutans* and *S. aureus*, which are well known as cariogenic bacteria [69,70,71]. In addition, CuBGn releases copper ions from nanoparticles even under bacterial infection and has effective antimicrobial activity for hard tissue repair [72,73], so it was expected to show higher antimicrobial activity when combined with SDF. Although the antimicrobial effect of SDF was less sensitive in *S. mutans*, it seemed clear that it increased with the addition of copper.

Adding CuBGn to SDF increased its viscosity, increasing the ease of its handling in the clinic by reducing leakage to the gingiva or pulp. This result leads to a decrease in cytotoxicity, thereby reducing the possibility of gingivitis or pulp necrosis. However, this addition should not reduce the capability of SDF to arrest and prevent caries, which is resolved because Cu increases the antibacterial activity against cariogenic organisms and Ag, F deposition on tooth surface.

## 5. Conclusions

In conclusion, SDF with CuBGn powder showed improved viscosity and flowability, which resulted in more fluoride and silver adhere on the HA surface, and showed excellent results on cell viability, and antibacterial effect without affecting fluoride releasement. Therefore, CuBGns are promising materials to be combined with various dental materials, and further investigation is needed to encourage the synergistic effect.

## Figures and Tables

**Figure 1 pharmaceutics-14-01137-f001:**
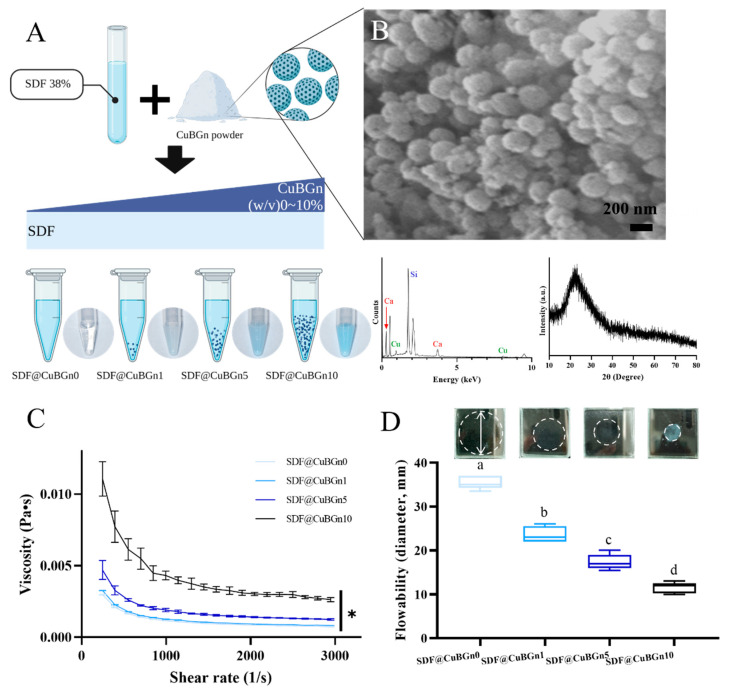
Preparation and characterization of copper-doped bioactive glass nanoparticles (CuBGns) and physical analysis of a silver diamine fluoride (SDF) solution mixed with CuBGns (SDF@CuBGns). (**A**) Schematic illustration of the preparation of SDF@CuBGn and a picture of the actual mix. CuBGn was placed in SDF for 0, 1, 5, and 10 (*w*/*v*)%, and the larger the amount of powder was, the more blue the solution. (**B**) Field emission scanning electron microscopy (FE-SEM) (50,000×, 5 kV), energy dispersive spectroscopy (EDS) and X-ray diffraction (XRD) of CuBGns. FE-SEM shows the numerous spherical morphologies of CuBGns, and EDS analysis confirmed the presence of elements Si, Cu and Ca. The presence of a broad halo (at 2θ = 20° to 30°) and absence of XRD diffraction peaks showed the amorphous structure. (**C**) Viscosity and (**D**) flowability of SDF@CuBGn. Viscosity was measured with 0.3 mL of sample placed between parallel plates (diameter 60.0 mm) (n = 3). The flow test was performed according to ISO 6876:2012. Then, 0.01 mL of SDF@CuBGn was placed onto a 40 mm × 40 mm, 5 mm thick glass plate (approximately 20 g) and covered with an identical glass plate. An additional 100 g of weight was placed on the sample and removed after 10 min. The maximum and minimum diameters of SDF@CuBGn were measured and averaged (n = 5). Adding more CuBGn resulted in a higher viscosity and lower flowability. There was a significant difference between groups with different letters (a, b, c, and d) in (**D**). Statistical analysis was performed using one-way analysis of variance (ANOVA) followed by Tukey’s post-hoc test. * *p* < 0.05.

**Figure 2 pharmaceutics-14-01137-f002:**
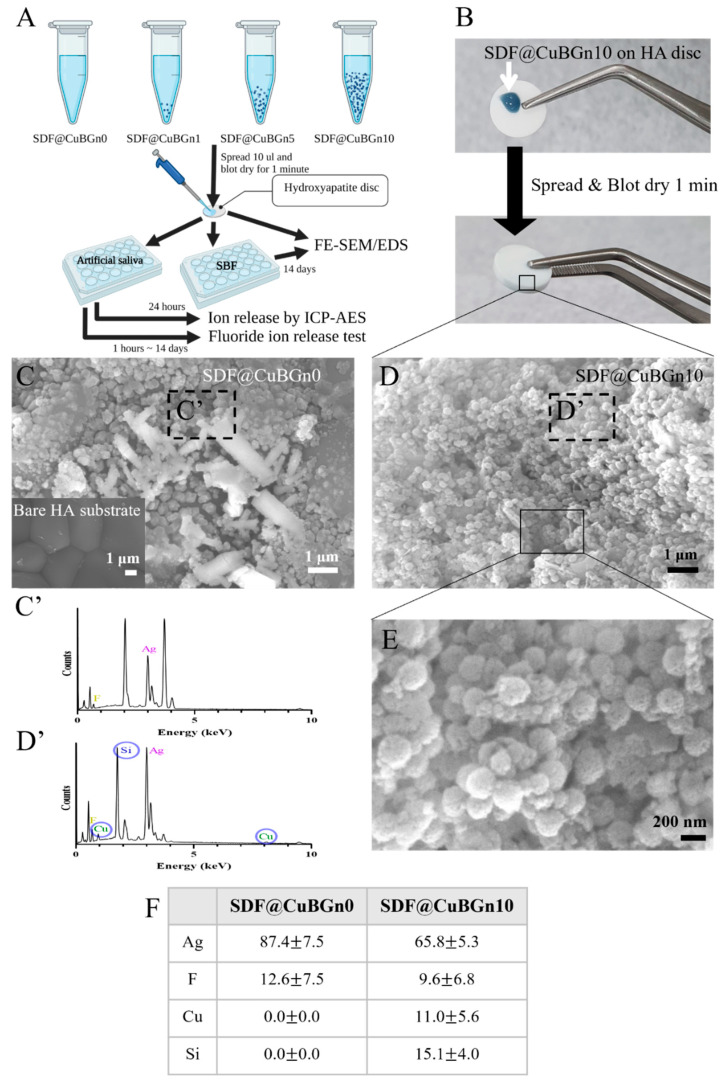
Physicochemical analysis (morphology and surface elemental analysis) of hydroxyapatite (HA) discs with SDF@CuBGn. (**A**) Schematic illustration of the physicochemical analysis of HA discs with SDF@CuBGn. Ten microliters of SDF@CuBGn were spread on HA discs (diameter 9.5 mm, thickness 1.8 mm) and blotted for 1 min. Discs were prepared and examined by FE-SEM and EDS after SDF@CuBGn was applied, immersed in simulated body fluid (SBF, at pH 7.4 and 37 °C) for 14 days, and ion release tests were performed after soaking in artificial saliva (37 °C, being agitated). (**B**) Actual picture of spreading SDF@CuBGn10 on HA discs. (**C**) HA disc with no treatment (bottom left) and SDF@CuBGn0, (**D**) SDF@CuBGn10 on HA disc analyzed with FE-SEM (10,000×, 20 kV) and (**C’**), (**D’**) EDS analysis of parts of (**C**) and (**D**). Ag and F were found in both samples, and Cu and Si were detected only in SDF@CuBGn10. (**E**) Magnified FE-SEM of SDF@CuBGn10 (40,000×, 20 kV). The spherical morphology of CuBGn, which was not seen in (**C**) SDF@CuBGn0, was found. (**F**) EDS analysis of components of SDF@CuBGn-applied HA discs (weight%).

**Figure 3 pharmaceutics-14-01137-f003:**
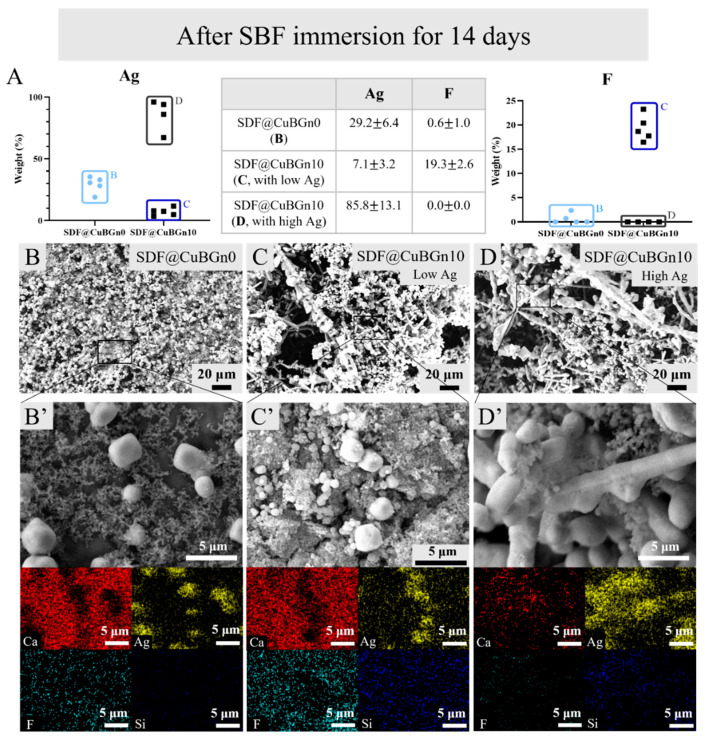
Physicochemical analysis (surface analysis after SBF immersion for 14 days) of SDF@CuBGn-applied HA discs. (**A**) Elemental EDS analysis of Ag and F. (**B**) SDF@CuBGn0-applied HA disc and (**B’**) magnified image. (**C**,**D**) SDF@CuBGn10-applied HA disc and (**C’**,**D’**) magnified image. SDF@CuBGn0- and SDF@CuBGn10-applied HA discs were immersed in SBF, which is optimal condition for accelerating remineralization. After 14 days, the samples were thoroughly washed, and the morphological change was observed by FE-SEM and EDS (200×, 5000×, 15 kV). SDF@CuBGn-applied discs showed many block-shape mass and EDS analysis revealed that they were Ag. Particularly, SDF@CuBGn10-applied disc showed that the silver blocks held together to form rod-shaped branches of Ag. The spot with lots of silver rods (**D**) showed very high silver content, while the other spots (**C**) showed high fluoride deposition. Adding CuBGns made silver-branch and even increased fluoride deposition in spots with low silver.

**Figure 4 pharmaceutics-14-01137-f004:**
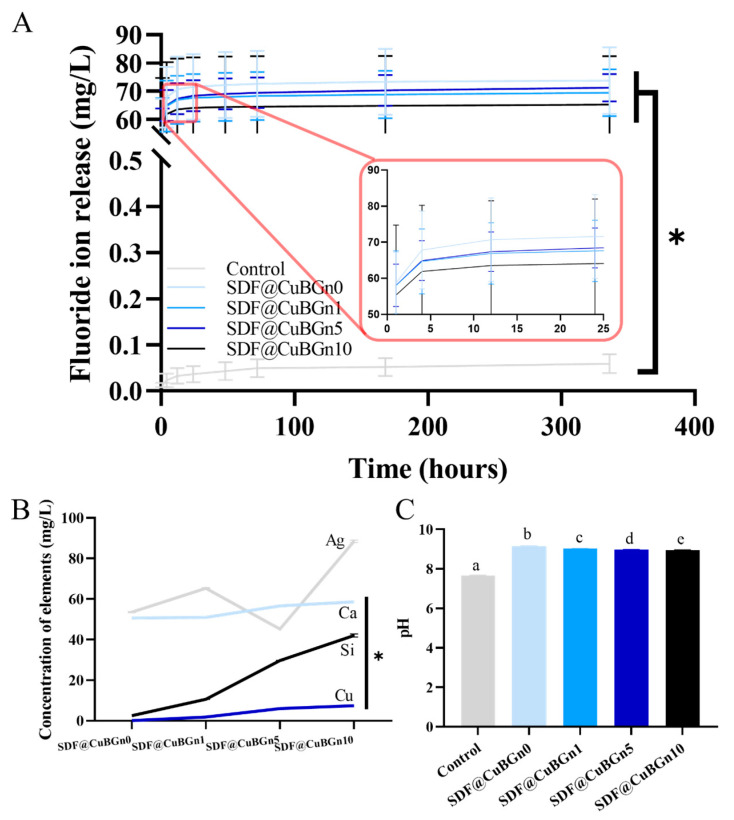
Physicochemical analysis (ion release tests for F, Ca, Cu, and Si and pH measurements) of SDF@CuBGn-applied HA discs. (**A**) Accumulated fluoride ion release test. More than 90% of fluoride was released within 4 h, and all test groups presented significantly higher (at least 1000 times) fluoride ion release than the control group (*p* < 0.05). CuBGn did not affect the result. (*p* > 0.05). (**B**) Ag, Ca, Cu, and Si ion concentrations and (**C**) pH measurements of artificial saliva with SDF@CuBGn-applied discs in it for 24 h. As the CuBGn increased, the concentration of released Ca, Cu, and Si ions also increased (*p* < 0.05) without affecting Ag ion release (*p* > 0.05), and the pH of the extracted media decreased, although it was higher than that of the control solution. There was a significant difference between groups with different letters (a, b, c, d, and e) in (**C**).Interestingly, adding CuBGn slightly decreased the pH despite hydrolysis of BGn producing OH^−^ ions, which may have been due to the amount of SDF@CuBGn attached to the HA disc increasing as the viscosity increased. Statistical analysis was performed using ANOVA and repeated-measures ANOVA only in the fluoride release test followed by Tukey’s post-hoc test. * *p* < 0.05, n = 3.

**Figure 5 pharmaceutics-14-01137-f005:**
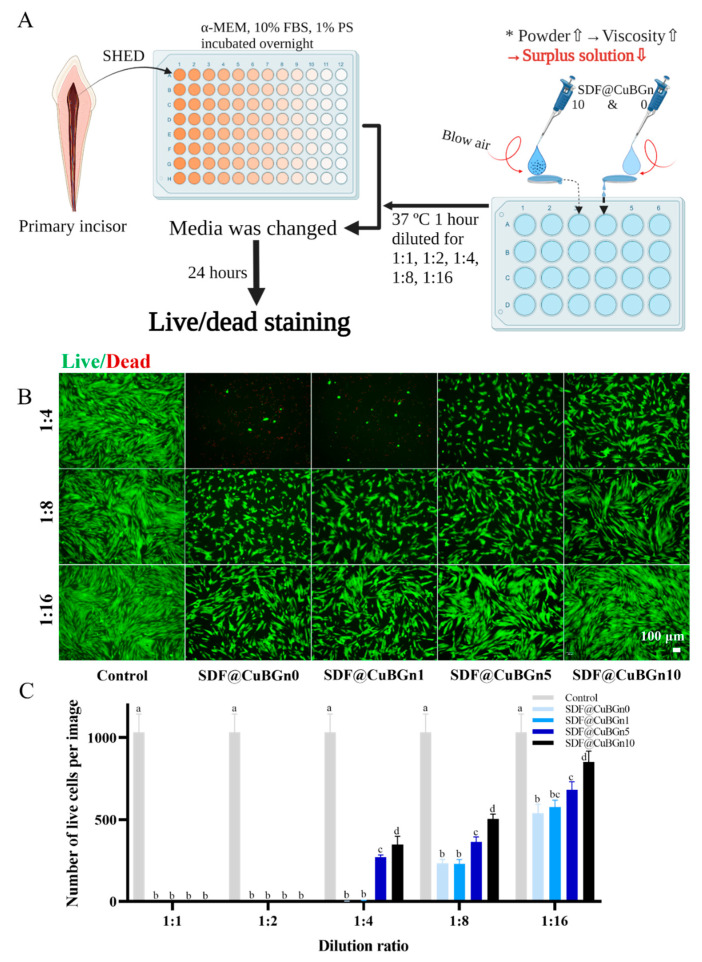
In vitro study of cytotoxicity on stem cells from human exfoliated deciduous teeth (SHED). (**A**) Schematic illustration of the cytotoxicity test of SDF@CuBGn on SHED. The higher the viscosity, the more solution was expected to attach to the HA disc. Based on this, the test was designed to simulate when SDF@CuBGn was applied to teeth. Ten microliters of SDF@CuBGn were placed on one end of the HA disc, and compressed air was blown to flow the solution to the other side. The surplus solution that fell into α-minimum essential medium (α-MEM) was used for the SHED cell viability test, and as expected, the relatively high-viscosity solution had less surplus solution. (**B**) Fluorescence image of live (green color)/dead (red color) cells. (**C**) Number of live cells per image. Surplus test solutions were diluted to 1:1, 1:2, 1:4, 1:8, and 1:16. At 1:1 and 1:2 dilution ratios in culture media, all groups of cells did not survive. At a 1:4 dilution ratio, SDF@CuBGn5 and SDF@CuBGn10 showed lower cell viability (approximately 30% compared to the control group), but both SDF@CuBGn0 and SDF@CuBGn1 had extremely low cell viability (<1% compared to the control group). At all dilution ratios, as the amount of powder increased, cell viability also increased due to the diminished surplus solution. Statistical analysis was performed using ANOVA followed by Tukey’s post-hoc test. There was a significant difference between groups with different letters (a, b, c, and d) in (**C**). * *p* < 0.05, n = 6.

**Figure 6 pharmaceutics-14-01137-f006:**
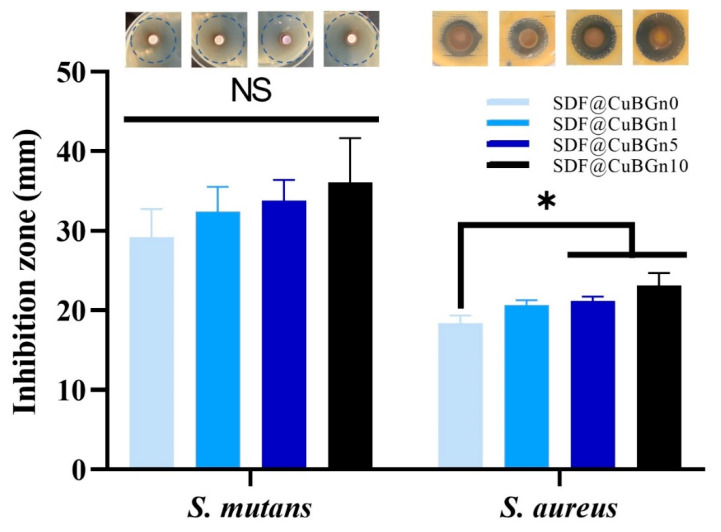
In vitro study of antibacterial effects on cariogenic organisms. To measure the antibacterial effect, agar diffusion tests were selected, which are simple, fast, and reliable. In *Streptococcus mutans* (*S. mutans*) and *Staphylococcus aureus* (*S. aureus*), the diameter of the inhibition zone increased as the powder increased, but it was not statistically significant in *S. mutans* (*p* > 0.05, expressed as NS). In *S. aureus*, SDF@CuBGn5 and SDF@CuBGn 10 clearly had greater effects than SDF@CuBGn0 (*p* < 0.05). Silver ions exhibit antibacterial effects, and in addition, CuBGn releases copper ions even under bacterial infection and has antimicrobial activity for hard tissue repair, so it was expected to show higher antimicrobial activity when it was combined with SDF. Statistical analysis was performed using ANOVA followed by Tukey’s post-hoc test. * *p* < 0.05, n = 3.

## Data Availability

Not applicable.

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
