# Peer review of "Characterization of Physical and Biological Properties of a Caries-Arresting Liquid Containing Copper Doped Bioglass Nanoparticles"

_pharmaceutics, 2022, doi:10.3390/pharmaceutics14061137_

Round 1

Reviewer 1 Report

I found the paper very nice and well written, reporting on an important topic. My only suggestion is that CuBGns are more characterised. Authors do not show XPS or XRD, for example. So more characterisation should be added, if possible, as the material is not well characterised in my opinion.

Moreover, please correct some minor mistypes such as analysisconfirmed (line 232), etc. In some cases the name of microorganisms are in italics, but in others (e.g. line 340), they are in plain text. The style should be uniformised.

Reviewer 2 Report

Paper addresses very important issues concerning SDF. Tooth discoloration and toxicity are the main problems in SDF application. The study is well designed and performed. Conclusions are based on the results and the discussion is comprehensive. My recommendation is to publish without any corrections.

Reviewer 3 Report

In this work copper-doped bioactive glass nanoparticles were combined with silver diamine fluoride to overcome several known problems of dental materials such as the high flowability due to low viscosity and cytotoxicity to the pulp. The work is interesting with considerable addition in the field and probably will earn wide interest which properties justify publication in Pharmaceutics. The references are upto date and comprehensive. The work is very nice and carefully discussed. I suggest publication after considering the following minor remarks:

The work itself is complete and no further examination is necessary. However one thing would be important to consider: these materials applied together with other, e.g. dental composites, which are photosensitive and the photo-polymerization is affected by the absorption and light scattering. The neutrality of materials examined here towards the applications of dental composites should be neutral or the possible effect should be known. I suggest authors to insert few sentences into their work about the possible interference with the applications together with the dental composites.    
